# *DenoiseRep*: Denoising Model for Representation Learning

**Zhengrui Xu**[1][†]
zrxu23@bjtu.edu.cn

**Guan'an Wang** [†‡]
guan.wang0706@gmail.com

**Xiaowen Huang** [1,2,3][*]
xwhuang@bjtu.edu.cn

**Jitao Sang** [1,2,3]
jtsang@bjtu.edu.cn

[1]**School of Computer Science and Technology, Beijing Jiaotong University**
[2]Beijing Key Lab of Traffic Data Analysis and Mining, Beijing Jiaotong University
[3]Key Laboratory of Big Data & Artificial Intelligence
in Transportation(Beijing Jiaotong University), Ministry of Education

## Abstract

The denoising model has been proven a powerful generative model but has little exploration of discriminative tasks. Representation learning is important in discriminative tasks, which is defined as *"learning representations (or features) of the data that make it easier to extract useful information when building classifiers or other predictors"* [4]. In this paper, we propose a novel Denoising Model for Representation Learning (*DenoiseRep*) to improve feature discrimination with joint feature extraction and denoising. *DenoiseRep* views each embedding layer in a backbone as a denoising layer, processing the cascaded embedding layers as if we are recursively denoise features step-by-step. This unifies the frameworks of feature extraction and denoising, where the former progressively embeds features from low-level to high-level, and the latter recursively denoises features step-by-step. After that, *DenoiseRep* fus es the parameters of feature extraction and denoising layers, and *theoretically demonstrates* its equivalence before and after the fusion, thus making feature denoising computation-free. *DenoiseRep* is a label-free algorithm that incrementally improves features but also complementary to the label if available. Experimental results on various discriminative vision tasks, including re-identification (Market-1501, DukeMTMC-reID, MSMT17, CUHK-03, vehicleID), image classification (ImageNet, UB200, Oxford-Pet, Flowers), object detection (COCO), image segmentation (ADE20K) show stability and impressive improvements. We also validate its effectiveness on the CNN (ResNet) and Transformer (ViT, Swin, Vmamda) architectures. Code is available at https://github.com/wangguanan/DenoiseRep.

## 1 Introduction

Denoising Diffusion Probabilistic Models (DDPM) [21] or Diffusion Model for short have been proven to be a powerful generative model [5]. Generative models can generate vivid samples (such as images, audio and video) by modeling the joint distribution of the data $P(X, Y)$, where $X$ is the sample and $Y$ is the condition. Diffusion models achieve this goal by adding Gaussian noise to the data and training a denoising model of inversion to predict the noise. Diffusion models can generate multi-formity and rich samples, such as Stable diffusion [50], DALL [47] series and Midjourney, these powerful image generation models, which are essentially diffusion models.

---

[†]Equal Contribution.
[‡]Project Lead.
[*]Corresponding Author.

38th Conference on Neural Information Processing Systems (NeurIPS 2024).

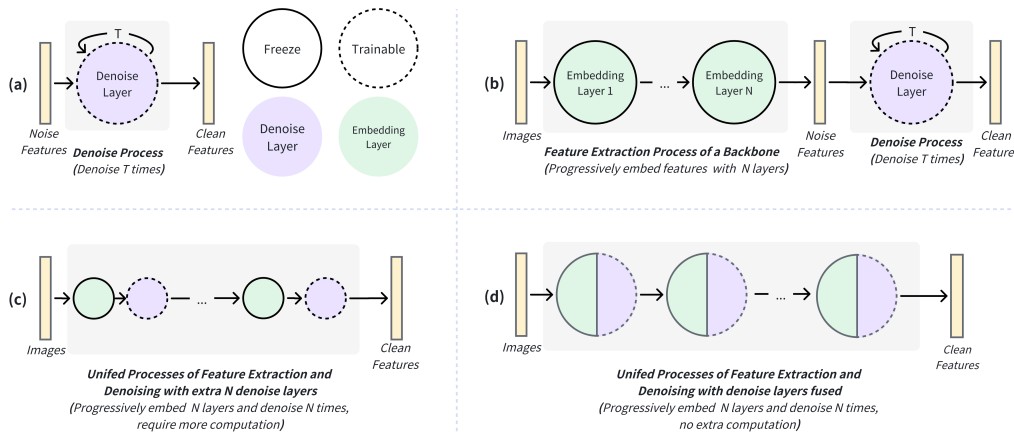

Figure 1: A brief description of our idea. (a) A typical denoising model for generative tasks recursively applies a denoising layer. (b) A naive idea that applies a denoising strategy to a discriminative model is applying a recursive denoise layer on the feature of a backbone and taking extra inference latency. (c,d) Our *DenoiseRep* first unifies the frameworks of feature extraction and denoising in a backbone pipeline, then merges parameters of the denoising layers into embedding layers, making the feature more discriminative without extra latency cost.

However, its application to discriminative models has not been extensively explored. Different from generative models, discriminative models predict data labels by modeling the marginal distribution of the data $P(Y|X)$. $Y$ can be various labels, such as image tags for classification, object boxes for detection, and pixel tags for segmentation. Currently, there are several methods based on diffusion models implemented in specific fields. For example, DiffusionDet [7] is a new object detection framework that models object detection as a denoising diffusion process from noise boxes to object boxes. It describes object detection as a generative denoising process and performs well compared to previous mature object detectors. DiffSeg [52] for image segmentation, which is a method of unsupervised zero-shot sample segmentation using pre-trained models (stable diffusion). It introduces a simple and effective iterative merging process to measure the attention maps between KL divergence and merge them into an effective segmentation mask. The proposed method does not require any training or language dependency to extract the quality segmentation of any image.

The methods above are carefully designed for specific tasks and require a particular data structure. For example, DiffusionDet [7] uses noise boxes and DiffSeg [52] uses noise segmentation. In this paper, we explore a more general conception of how the denoising model can improve representation learning, *i.e. "learning representations (or features) of the data that make it easier to extract useful information when building classifiers or other predictors"* [4], and contribute to discriminative models. We take person Re-Identification (ReID) [66, 3] as a benchmark task. ReID aims to match images of a pedestrian under disjoint cameras, and is suffered by pose, lighting, occlusion and so on, thus requiring more identity-discriminative feature.

A straightforward approach is applying the denoising process to a backbone's final feature [26, 14], reducing noise in the final output and making the feature more discriminative, as Fig. 1(b) shows. However, this way can be computationally intensive. Because the denoising layer needs to be proceeded on the output of the previous one in a recursive and step-by-step manner. Considering that a backbone typically consists of cascaded embedding layers (e.g., convolution layer, multi-head attention layer), we propose a novel perspective: treating each embedding layer as a denoising layer. As shown in Fig. 1(c), it allows us to process the cascaded layers as if we are recursively proceeding through the denoising layer step-by-step. This method transforms the backbone into a series of denoising layers, each working on a different feature extraction level. While this idea is intuitive and simple, its practical implementation presents a significant challenge. The main issue arises from the requirement of the denoising layer for the input and output features to exist in the same feature space. However, in a typical backbone (*e.g.* ResNet [26], ViT [14])), the layers progressively map features

from a low level to a high level. It means that the feature space changes from layer to layer, which contradicts the requirement of the denoising layer.

To resolve all the difficulties above and efficiently apply the denoising process to improve discriminative tasks, our proposed Denoising Model for Representation Learning (*DenoiseRep*) is as below: Firstly, we utilize a well-trained backbone and keep it fixed throughout all subsequent procedures. This step is a free launch as we can easily use any publicly available backbone without requiring additional training time. This approach allows us to preserve the backbone's inherent characteristics of semantic feature extraction. Given the backbone and an image, we can get a list of features. Next, we train denoising layers on those features. The weights of denoising layers are randomly initialized and their weights are not shared. The training process is the same as that in DDPM [21], where the only difference is that the denoising layer in DDPM takes a dynamic $t \in [1, T]$, and our denoising layers take fixed $n \in [1, N]$, where $n$ is the layer index, $T$ is denoise times and $N$ is backbone layer number as shown in Fig. 1(c). Finally, considering that the $N$ denoising layers consume additional execution latency, we propose a novel feature extraction and feature denoising fusion algorithm. As shown in Fig. 1(d), the algorithm merges parameters of extra denoising layers into weights of the existing embedding layers, thus enabling joint feature extraction and denoising without any extra computation cost. We also *theoretically demonstrate* the total equivalence before and after parameter fusion. Please see Section 3.3 and Eq (7) for more details.

Our contributions can be summarized as follows:

(1) We propose a novel Denoising Model for Representation Learning (*DenoiseRep*), which innovatively integrates the denoising process, originating from generative tasks, into the discriminative tasks. It treats $N$ cascaded embedding layers of a backbone as $T$ times recursively proceeded denoising layers. This idea enables joint feature extraction and denoising is a backbone, thus making features more discriminative.

(2) The proposed *DenoiseRep* fuses the parameters of the denoising layers into the parameters of the corresponding embedding layers and *theoretically* demonstrates their equivalence. This contributes to a computation-efficient algorithm, which takes no extra latency.

(3) Extensive experiments on 4 ReID datasets verified that our proposed *DenoiseRep* can effectively improve feature performance in a label-free manner and performs better in the case of label-argumented supervised training or introduction of additional training data. We also extend *DenoiseRep* to large-scale (ImageNet), fine-grained (CUB200, Oxford-Pet, Flowers) image classifications, object detection (COCO) and image segmentation (ADE20K), showing its scalability.

## 2  Related Work

**Generative models** learn the distribution of inputs before estimating class probabilities. A generative model learns the data generation process by learning the probability distribution of the input data and generating new data samples. The generative models first estimate the conditional density of categories $P(x|y = k)$ and prior category probabilities $P(y = k)$ from the training data. The $P(x)$ is obtained by the full probability formula. So as to model the probability distribution of each type of data. Generative models can generate new samples by modelling data distribution. For example, Generative Adversarial Networks (GANs) [17, 43, 23] and Variational Autoencoders (VAEs) [24, 53, 69, 64] are both classic generative models that generate real samples by learning potential representations of data distributions, demonstrating excellent performance in data distribution modeling. Recent research has focused on using **diffusion models** for generative tasks. The diffusion model was first proposed by the article [51] in 2015, with the aim of eliminating Gaussian noise from continuous application to training images. The DDPM [21] proposed in 2020 have made the use of diffusion models for image generation mainstream. In addition to its powerful generation ability, the diffusion model also has good denoising ability through noise sampling, which can denoise noisy data and restore its original data distribution.

**Discriminative models** learn condition distribution, *i.e.* $P(y|x)$, where $x$ is data and $y$ is task-specific features. For example, classification tasks [1, 2, 13] map data to tags, retrieval tasks [36, 62] map data to a feature space where similar data should be near otherwise faraway, detection task [49, 27] map data to space position and size. **Person Re-Identification (ReID)** is a fine-grained retrieval task which identifies individuals among disjoint camera views. Considering its challenge to feature

discrimination, we take ReID as the major benchmark task and the others as auxiliary benchmarks. Existing ReID methods can be grouped into hand-crafted descriptors [35, 42, 65] incorporated with metric learning [25, 34, 71] and deep learning algorithms [58, 57, 56, 44, 16, 10]. State-of-the-art ReID models often leverage convolutional neural networks (CNNs) [28] to capture intricate spatial relationships and hierarchical features within person images. Attention mechanisms [54, 14], spatial-temporal modeling [31, 30], and domain adaptation techniques [9] have further enhanced the adaptability of ReID models to diverse and challenging real-world scenarios.

## 3 *DenoiseRep*: Denoising Model for Representation Learning

### 3.1 Review Representation Learning

Representation learning plays a pivotal role in discriminative tasks, which is defined as *"learning representations (or features) of the data that make it easier to extract useful information when building classifiers or other predictors"* [4]. A common architecture of discriminative tasks consists of a vision backbone to extract discriminative features (*e.g.*, ResNet [18], ViT [14]) and a task-specific head that operates on these features (*e.g.* MLP [26] for classification, RCNN [49] for object detection, FCN [40] for segmentation). It is evident that the vision backbone is central to representation learning. In this paper, we introduce a novel Denoising Model for Representation Learning (*DenoiseRep*), which integrates feature extraction and feature denoising within a single vision backbone. This approach aims to enhance the discriminative power of the features extracted.

### 3.2 Joint Feature Extraction and Feature Denoising

We refer to the diffusion modeling approach to denoise the noisy features through T-steps to obtain clean features. At the beginning, we use the features output from the backbone network as data samples for diffusion training, and get the noisy samples by continuously adding noise and learning through the network in order to simulate the data distribution of its features.

$$q(\mathbf{x}_{1:T}|\mathbf{x}_0) := \prod_{t=1}^{T} q(\mathbf{x}_t|\mathbf{x}_{t-1}) \tag{1}$$

$$q(\mathbf{x}_t|\mathbf{x}_{t-1}) := \mathcal{N}(\mathbf{x}_t; \sqrt{1-\beta_t}\mathbf{x}_{t-1}, \beta_t\mathbf{I}) \tag{2}$$

where $X_0$ represents the feature vector output by the backbone, $t$ represents the diffusion step size, $\beta_t$ is a set of pre-set parameters, and $X_t$ represents the noise sample obtained through diffusion process.

In the inference stage, as shown in Fig. 1(b), we perform T-step denoising on the output features, to obtain cleaner features and improve the expressiveness of the features.

$$p_\theta(\mathbf{x}_{0:T}) := p(\mathbf{x}_T) \prod_{t=1}^{T} p_\theta(\mathbf{x}_{t-1}|\mathbf{x}_t) \tag{3}$$

$$p_\theta(\mathbf{x}_{t-1}|\mathbf{x}_t) := \mathcal{N}(\mathbf{x}_{t-1}; \mu_\theta(\mathbf{x}_t, t), \Sigma_\theta(\mathbf{x}_t, t)) \tag{4}$$

where $X_t$ represents the feature vector output by the backbone in the inference stage. $T$ is the denoising step size, representing the magnitude of the noise. We adjust $t$ appropriately based on different datasets and backbones to obtain the optimal denoising amplitude. According to $p_\theta(\mathbf{x}_{t-1}|\mathbf{x}_t)$ denoise it step by step, and finally obtains $X_0$, which represents the clean feature after denoising.

### 3.3 Fuse Feature Extraction and Feature Denoising

As described in Section 3.2, the proposed method above could effectively improve the discriminability of features. Still, extra inference latency is introduced caused by recursive calling of the denoising layers. To solve the problem, we propose to fuse parameters of feature denoising layers into parameters of existing embedding layers of the backbone. The core idea is to expand the linear layer each transformer encoder block into two branches, one for its original embedding layer and the other for extra denoising layer. As shown in Fig. 2, during the training phase, we freeze the original embedding layers and only train the denoising layers. The training method is consistent with section 3.2, and the features are diffused and fed into the denoising layers. Please refer to Algorithm 1 for

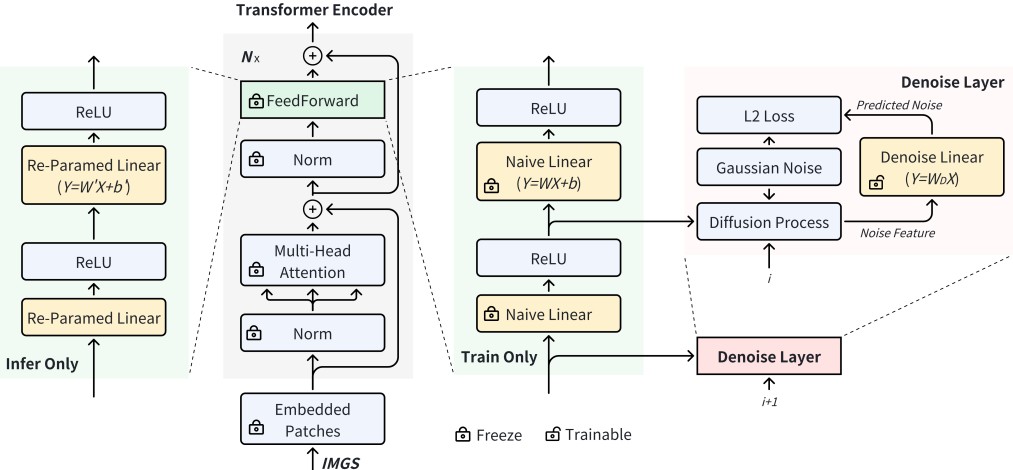

Figure 2: Pipeline of our proposed *DenoiseRep*. ViT consists of $N$ cascaded transformer encoder layers. During the training phase (see the right side "Train Only" process), we freeze the backbone parameters and only train the extra denoising layers. In the inference stage (see the left side "Infer Only" process), we merge the parameters of denoising layers to corresponding encoder layers. So there is no extra inference latency cost. Please find definitions of $W$, $b$, $W_D$, $W'$, $i$ and $b'$ in Algorithm 2.

more details. In the inference stage, we fuse the pre-trained parameters of embedding and denoising layers, merging the two branches into a single branch without additional inference time. Please note that, here we take the transformer architecture as an example, but *DenoiseRep* is suitable for CNN architecture. We demonstrate its scalbility on CNN in experiments. The derivation of parameter merging is as follows:

$$X_{t-1} = \frac{1}{\sqrt{a_t}}(X_t - \frac{1-a_t}{\sqrt{1-\bar{a}_t}}D_\theta(X_t, t)) + \sigma_t z \qquad (5)$$

where $a_t = 1 - \beta_t$, $D_\theta$ are the parameters of the prediction noise network.

$$Y = WX + b$$

$$\frac{1}{\sqrt{a_t}}X_t - X_{t-1} = \frac{1-a_t}{\sqrt{a_t}\sqrt{1-\bar{a}_t}}D_\theta X_t - \sigma_t z \qquad (6)$$

$$\frac{1}{\sqrt{a_t}}Y_t - Y_{t-1} = \frac{1-a_t}{\sqrt{a_t}\sqrt{1-\bar{a}_t}}D_\theta Y_t - \sigma_t z$$

We make a simple transformation of Eq. (5) and multiply both sides simultaneously by $W$. The simplified equation can be obtained by bringing $Y_t$ in terms of $WX_t + b$:

$$Y_{t-1} = [W - C_1(t)WW_D]X_t + WC_2(t)C_3 + b$$

$$C_1(t) = \frac{1-a_t}{\sqrt{a_t}\sqrt{1-\bar{a}_t}} \qquad C_2(t) = \frac{1-a_{t-1}^-}{1-\bar{a}_t}\beta_t \qquad C_3 = Z \sim N(0, I) \qquad (7)$$

where $W_D$ denotes the parameters of $D_\theta(X_t, t)$, $X_t$ denotes the input of this linear layer, $Y_t$ denotes the output of this linear layer, and $Y_{t-1}$ denotes the result after denoising in one step of $Y_t$. Due to the cascading relationship of blocks, as detailed in Algorithm. 2, different $t$ values are set according to the order between levels, and the one-step denoising of one layer is combined to achieve the denoising process of $Y_t \to Y_0$, ensuring the continuity of denoising and ultimately obtaining clean features. We split the original single branch into a dual branch structure. During the training phase, the backbone maintains its original parameters and needs to train the denoising module parameters. In the inference stage, as shown on the left side of Fig. 2, we use the method of reparameterization, to replace the original $W$ parameter with $W'$, where $W' = [W - C_1(t)WW_D]$ in Eq. (7), which has the same

---

**Algorithm 1** Training

---

**Input:** The number of feature layers in the backbone N, features extracted from each layer $\{F_i\}_{i=1}^N$, the denoising module that needs to be trained $\{D_i(\cdot)\}_{i=1}^N$.

1: **repeat**
2:     **for** each $i \in [N, 1]$ **do**
3:         $t = i$: Specify the diffusion step t for the current layer based on the order of layers.
4:         $\epsilon \sim N(0, I)$: Randomly sample a Gaussian noise.
5:         $X_t = \sqrt{\bar{a}_t}F_i + \sqrt{1 - \bar{a}_t}\epsilon$: Forward diffusion process in Eq.(2).
6:         Take gradient descent step on $\nabla_\theta \|\epsilon - D_i(X_t, t)\|^2$
7:     **end for**
8: **until** converged

---

number of parameters as $W$, thus achieving the combination of $FC$ operation and denoising without additional time cost. It is a **Computation-free** method.

In Eq. (7), we achieve one-step denoising $Y_t \to Y_{t-1}$. If we need to increase the denoising amplitude, we can extend it to two-step or multi-step denoising. The following is the derivation formula for two-step denoising:

$$\frac{1}{\sqrt{a_t}}Y_t - Y_{t-1} = C_1(t)D_\theta Y_t - \sigma_t z \tag{8}$$

$$\frac{1}{\sqrt{a_{t-1}}}Y_{t-1} - Y_{t-2} = C_1(t-1)D_\theta Y_{t-1} - \sigma_{t-1}z \tag{9}$$

We can obtain this by eliminating $Y_{t-1}$ from Eq.(8) and Eq.(9) and replacing $Y_t$ with $WX_t + b$:

$$Y_{t-2} = W''X_t + C''$$

$$W'' = \frac{1}{\sqrt{a_t - 1}}\{\frac{W}{\sqrt{a_t}} - [C_1(t) + C_1(t-1)]WW_D + \sqrt{a_t}C_1(t-1)C_1(t)WW_DW_D\} \tag{10}$$

$$C'' = \frac{1}{\sqrt{a_t - 1}}[WC_2(t) + \sqrt{a_t}WC_2(t-1) - \sqrt{a_t}C1_{(t-1)}C_2(t)WW_D]Z + b$$

Note that a single module completes two steps of denoising. To ensure the continuity of denoising, the $t$ value should be sequentially reduced by 2.

Our proposed *DenoiseRep* is based on feature-level denoising and can be migrated to various downstream tasks. It denoises the features on each layer for better removal of noise at each stage, as the noise in the inference stage comes from multiple sources, which could be noise in the input image or noise generated while passing through the network. Denoising each layer avoids noise accumulation and gives better quality output. And according to the noise challenges brought by data in different scenarios, the denoising intensity can be adjusted by controlling $t$, $\beta_t$, and the number of denoising times, which has good generalization ability.

---

**Algorithm 2** Inference

---

**Input:** The number of feature layers in the backbone N, features extracted from each layer $\{F_i\}_{i=1}^N$, trained denoising module parameters $\{W_{D_i}\}_{i=1}^N$ in Algorithm(1), after obtaining the initial feature $F^N$ through patch_embed, it is necessary to remove N-step noise from it, the pre-trained parameters $\{W_i\}_{i=1}^N$ and $\{b_i\}_{i=1}^N$ for the backbone.
**Output:** Feature $F^0$ after denoising.

1: **for** each $i \in [N, 1]$ **do**
2:     $t = i$: Set the denoising amplitude based on the depth of the current layer.
3:     $W' = [W_i - C_1(t)W_iW_{D_i}]$, $b' = W_iC_2(t)C_3 + b_i$: Parameter fusion according to Eq.(7).
4:     $F^{t-1} = W'F^t + b'$: Fuse feature extraction and feature denoising.
5: **end for**
6: **return** $F^0$

---

### 3.4 Unsupervised Learning Manner

Our proposed *DenoiseRep* is label-free because its essence is a generative model that models data by learning its distribution. Thus the training loss contains only the $Loss_p$ of denoising layers:

$$Loss_p = \sum_{i=1}^{N} |\epsilon_i - D_{\theta_i}(X_{t_i}, t_i)| \tag{11}$$

where $\epsilon$ denotes the sampled noise, $N$ denotes the number of denoising layers, $X_t$ denotes the noise sample, $t$ denotes the diffusion step, and $D_\theta(X_t, t)$ denotes the noise predicted by the denoising layer.

However, it is worth noting that our method is complementary to label if the label is available. $Loss_l$ is the task-specific supervised loss with label, $\lambda$ is the trade-off parameter between two losses. The label-argumented learning is defined as:

$$Loss = (1 - \lambda)Loss_l + \lambda Loss_p \tag{12}$$

Results in experiments Section 4.1 shows the improvement from label.

## 4 Experiments

Table 1: Experimental results on various discriminative tasks.

| Task | Model | Backbone | Dataset | Metric | *Baseline* | *+DenoiseRep* |
|---|---|---|---|---|---|---|
| Classification | SwinT [39] | SwinV2-T | ImageNet-1k | acc@1 | 81.82% | 82.13% |
| Person-ReID | TransReID-SSL [41] | ViT-S | MSMT17 | mAP | 66.30% | 67.33% |
| Detection | Mask-RCNN [19] | Swin-T | COCO | AP | 42.80% | 44.30% |
| Segmentation | FCN [40] | ResNet-50 | ADE20K | BIoU | 28.70% | 29.90% |

Our proposed *DenoiseRep* is a versatile method that can be incrementally applied to various discriminative tasks. Table 1 demonstrates that *DenoiseRep* yields stable and substantial improvements across image classification, object detection, image segmentation, and person re-identification. Given that person re-identification is a nuanced image retrieval task that poses a greater challenge to feature discriminability, we take it as our benchmark for model analysis. Details of the experimental settings are provided in Appendix A. Additional experimental results on various tasks are presented in Appendices B, C, D, and E.

### 4.1 Analysis of Label Informations

Table 2: *DenoiseRep* is a label-free method that can also be effectively complemented with labels when they are available. The table below analyzes the effectiveness of using labels. The baseline method, TransReID-SSL, is based on a ViT-small backbone. "Label-free" indicates training without labels, "label-augmented" refers to the use of labels, and "merged dataset" denotes the use of combined datasets without labels.

| Method | DukeMTMC(%) | MSMT17(%) | Market1501(%) | CUHK-03(%) |
|---|---|---|---|---|
| TransReID-SSL | 81.20 | 66.30 | 91.20 | 83.50 |
| *+DenoiseRep* (label-free) | 81.72 (↑ 0.52) | 66.87 (↑ 0.57) | 91.82 (↑ 0.62) | 83.72 (↑ 0.22) |
| *+DenoiseRep* (label-aug) | 82.12 (↑ 0.92) | 67.33 (↑ 1.03) | 92.05 (↑ 0.85) | 84.11 (↑ 0.61) |
| *+DenoiseRep* (merged ds) | 81.78 (↑ 0.58) | 66.99 (↑ 0.69) | 91.80 (↑ 0.60) | 83.86 (↑ 0.36) |

As mentioned in Section 3.3, *DenoiseRep* is an unsupervised denoising module, and its training does not require the assistance of label information. We conducte the following experiments to identify three key issues.

(1) *Is this label-free and unsupervised training denoising plugin effective?* As shown in Table 2 (line2), compared with baseline method (line1), the baseline method performs better after adding our **label-free** plugin, which shows that our method does have denoising capability for features.

(2) *Could introducing label information for supervised training further improve performance?* Introducing label information is actually adding $Loss_l$ as mentioned in Section 3.4 as a supervised signal. As shown in Table 2 (line3), baseline method with label-argumented *DenoiseRep* achieve improvements of 0.32% - 0.70% on the mAP metric, indicating that our denoising plugin has label compatibility, in other words, the plug-in is effective for feature denoising regardless of label-argumented supervised or lable-free unsupervised training.

(3) *Since our plugin can perform unsupervised denoising of features, it is natural to think about whether adding more data for training the plugin could further improve its performance?* We merge four datasets for training and then test on each dataset using mAP to evaluate. Comparing the results of training on sigle dataset (line2) with on merged datasets (line4), we found that adopting other datasets for unsupervised learning can further improve the performance of *DenoiseRep*, which also proves that *DenoiseRep* has good generalization ability.

To demonstrate that our method can perform unsupervised learning and has good generalization, we merged four datasets and rearranged the sequence IDs to ensure the reliability of the experiment. The model is tested on the entire dataset. During the training process, we freeze the baseline parameters and only train the *DenoiseRep* module, without the need for labels, for unsupervised learning. Then test on a single dataset and compare the results of training on a single dataset. As shown in Table 2, it can be observed that adding unlabeled training data from different datasets can improve the model's performance on a single dataset, proving that this module has a certain degree of generalization.

## 4.2   Comparison with State-of-the-Art ReID Methods

We compare several state-of-the-art ReID methods on four datasets. One of the best performing comparison methods is TransReID-SSL, which is a series of ReID methods based on the ViT backbones. Other methods are based on structures such as CNNs. We add our method to TransReID-SSL series and observe their performance. As shown in Table 3, we have the following findings:

Table 3: Comparison with state-of-the-art ReID methods.

| Method | Backbone | MSMT17 | | Market1501 | | DukeMTMC | | CUHK03-L | |
|---|---|---|---|---|---|---|---|---|---|
| | | mAP | R1 | mAP | R1 | mAP | R1 | mAP | R1 |
| MGN [59] | ResNet-50 | – | – | 86.90 | 95.70 | 78.40 | 88.70 | 67.40 | 68.00 |
| OSNet [74] | OSNet | 52.90 | 78.70 | 84.90 | 94.80 | 73.50 | 88.60 | – | – |
| BAT-net [15] | GoogLeNet | 56.80 | 79.50 | 87.40 | 95.10 | 77.30 | 87.70 | 76.10 | 78.60 |
| ABD-Net [8] | ResNet-50 | 60.80 | 82.30 | 88.30 | 95.60 | 78.60 | 89.00 | – | – |
| RGA-SC [68] | ResNet-50 | 57.50 | 80.30 | 88.40 | 96.10 | – | – | 77.40 | 81.10 |
| ISP [76] | HRNet-W32 | – | – | 88.60 | 95.30 | 80.00 | 89.60 | 74.10 | 76.50 |
| CDNet [29] | CDNet | 54.70 | 78.90 | 86.00 | 95.10 | 76.80 | 88.60 | – | – |
| Nformer [60] | ResNet-50 | 59.80 | 77.30 | 91.10 | 94.70 | 83.50 | 89.40 | 78.00 | 77.20 |
| TransReID [20] | ViT-base-ics | 67.70 | 85.30 | 89.00 | 95.10 | 82.20 | 90.70 | 84.10 | 86.40 |
| TransReID | ViT-base | 61.80 | 81.80 | 87.10 | 94.60 | 79.60 | 89.00 | 82.30 | 84.60 |
| TransReID-SSL [41] | ViT-small | 66.30 | 84.80 | 91.20 | 95.80 | 81.20 | 87.80 | 83.50 | 85.90 |
| TransReID-SSL | ViT-base | 75.00 | 89.50 | 93.10 | 96.52 | 84.10 | 92.60 | 87.80 | 89.20 |
| CLIP-REID [32] | ViT-base | 75.80 | 89.70 | 90.50 | 95.40 | 83.10 | 90.80 | – | – |
| TransReID + *DenoiseRep* | ViT-base-ics | 68.10 | 85.72 | 89.56 | 95.50 | 82.35 | 90.87 | 84.15 | 86.39 |
| TransReID + *DenoiseRep* | ViT-base | 62.23 | 82.02 | 87.25 | 94.63 | 80.12 | 89.33 | 82.44 | 84.61 |
| TransReID-SSL + *DenoiseRep* | ViT-small | 67.33 | 85.50 | 92.05 | 96.68 | 82.12 | 88.72 | 84.11 | 86.47 |
| TransReID-SSL + *DenoiseRep* | ViT-base | 75.35 | 89.62 | **93.26** | **96.55** | **84.31** | **92.90** | **88.08** | **89.29** |
| CLIP-REID + *DenoiseRep* | ViT-base | **76.30** | **90.60** | 91.10 | 95.80 | 83.70 | 91.60 | – | – |

(1) Our method stands out on four datasets on ViT-base backbone with a large number of parameters, achieving almost the best performance on two evaluation metrics.

(2) The methods using our plugin outperforms the original methods with the same backbone on all datasets. In addition, the performance improvement of small-scale backbones with the addition of *DenoiseRep* is more significant than the large-scale backbones approach due to the fact that *DenoiseRep* is essentially a denoising module that removes the noise contained in the features during

the inference stage. For large-scale backbones, the extracted features already have good performance, so the denoising amplitude is limited. It has already fitted the dataset well. For small-scale backbones with poor performance, due to their limited fitting ability, there is a certain amount of noise in the extracted features during the inference stage. Denoising them can obtain better feedback.

(3) In fact, our method can be applied to any other backbone, just add it to each layer. In particular, the performance improvement of adding the denoising plugin to a poorly performing backbone might be more significant. This needs to be further verified in subsequent work. However, it is undeniable that we have verified the denoising ability of the *DenoiseRep* in the currently optimal ReID method.

In this section, a comparative analysis was conducted on four datasets to assess various existing ReID methods. These methods represent current mainstream ReID approaches, employing ResNet101, ViT-S, ViT-B, and ResNet50 as backbone architectures for feature extraction, respectively. Experimental results indicate that our proposed method outperforms other approaches in terms of both mAP and Rank-1 metrics.

## 4.3 Analysis of Parameter Fusion

The proposed *DenoiseRep* is computation-free. In section 3.3, we proved by theoretical derivation that inserting our denoising layer into each feature layer and fusion it does not introduce additional computation. In this section, we also conduct related validation experiments, the results of which are shown in Table 4.

Table 4: Parameter Fusion Performance Analysis. The *DenoiseRep$^-$* denoises based on the features of the final layer, while the *DenoiseRep* denoises based on the features of each layer. The baseline method TransReID-SSL is based on ViT-small backbone.

| Method | DukeMTMC | MSMT17 | Market1501 | CUHK-03 | Inference Time |
|---|---|---|---|---|---|
| TransReID-SSL | 81.20% | 66.30% | 91.20% | 83.50% | **0.34s** |
| *+DenoiseRep$^-$* | 81.56% | 66.81% | 91.07% | 83.59% | 0.39s (+15%) |
| *+DenoiseRep* | 82.12% | 67.33% | 92.05% | 84.11% | **0.34s** (+0%) |

Compare to the baseline method TransReID-SSL, adding *DenoiseRep$^-$* is able to improve the the performance, proving that feature based denoising is effective. However, it also brings extra inference latency (about 15%) because it is adding an extra parameter-independent denoising module at the end of the model.

Adopting *DenoiseRep* achieves a greater increase, it denoise the features on each layer, which can better remove noise at each stage because the noise in the inference stage comes from multiple aspects, which may be the noise in the input image or generated when passing through the network. Denoising each layer avoids noise accumulation and obtains a better quality output. Most importantly, since the operation of fusion can merge the parameters of the denoising module with the original parameters, the adoption of *DenoiseRep* does not take extra inference latency cost, which is a **computation-free** efficient approach.

## 4.4 Experiments on Classification Tasks

The *DenoiseRep* is based on denoising at the feature level and demonstrates strong generalization capabilities. To validate this generalization ability, we conduct experiments on other vision tasks to test the effectiveness of the *DenoiseRep*. We validate the generalization ability of *DenoiseRep* in image classification tasks on ImageNet-1k [12] datasets and three fine-grained image classification datasets (CUB200 [55], Oxford-Pet [46], and Flowers [45]). The accuracy index is chosen as the evaluation metric to assess model performance.

As shown in Table 5, we compare multiple classic backbones for representation learning on ImageNet-1k, and after adding the *DenoiseRep*, the accuracy of both top-1 and top-5 metrics improve without adding model parameters. Our method shows significant improvement in accuracy metrics compared to baseline on three fine-grained classification datasets. Prove that the *DenoiseRep* can improve the model's ability in image classification for different classification tasks. Additionally, our method proves to enhance the model's representation learning ability and extract more effective features

Table 5: The effectiveness of our method in image classification tasks was validated on three fine-grained classification datasets (CUB200, Oxford-Pet, Flowers) and ImageNet-1k.

| Method | Datasets | Param | acc@1 | | acc@5 | |
| --- | --- | --- | --- | --- | --- | --- |
| | | | *Baseline* | *+DenoiseRep* | *Baseline* | *+DenoiseRep* |
| SwinV2-T [39] | ImageNet-1k | 28M | 81.82% | 82.13% | 95.88% | 96.06% |
| Vmanba-T [38] | ImageNet-1k | 30M | 82.38% | 82.51% | 95.80% | 95.89% |
| ResNet50 [18] | ImageNet-1k | 26M | 76.13% | 76.28% | 92.86% | 92.95% |
| ViT-B [14] | CUB200 | 87M | 91.78% | 91.99% | – | – |
| ViT-B | Oxford-Pet | 87M | 94.37% | 94.58% | – | – |
| ViT-B | Flowers | 87M | 99.12% | 99.30% | – | – |

through denoising without incurring additional time costs. More experimental analysis can be found in Table 7 in Section C of the appendix.

## 5 Conclusion

In this work, we demonstrate that the diffusion model paradigm is effective for feature level denoising in discriminative model, and propose a computation-free and label-free method: *DenoiseRep*. It utilizes the denoising ability of diffusion models to denoise the features in the feature extraction layer, and fuses the parameters of the denoising layer and the feature extraction layer, further improving retrieval accuracy without incurring additional computational costs. We validate the effectiveness of the *DenoiseRep* method on multiple common image discrimination task datasets.

## Acknowledgement

This work was supported by the National Natural Science Foundation of China (62202041), National Key Research and Development Program of China under Grant (2023YFC3310700) and Fundamental Research Funds for the Central Universities (2023JBMC057).

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

## A  Experimental settings

**Datasets and Evaluation metrics**. We conduct training and evaluation on four datasets: DukeMTMC-reID [72], Market-1501 [70], MSMT17 [61], and CUHK-03 [33]. These datasets encompass a wide range of scenarios for person re-identification. For accuracy, we use standard metrics including Rank-1 curves (The probability that the image with the highest confidence in the search results is the correct result.) and mean average precision (MAP). All the results are from a single query setting.

**Implementation Details**. We implement our method using Python on a server equipped with a 2.10GHz Intel Core Xeon (R) Gold 5218R processor and two NVIDIA RTX 3090 GPUs. The epochs we trained are set to 120, the learning rate is set to 0.0004, the batch size during training is 64, the inference stage is 256, and the diffusion step size $t$ is set to 1000.

**Training and evaluation**. To better constrain the performance of the denoised features of the *DenoiseRep* for downstream tasks, we employ alternating fine-tune methods. The parameters of the *DenoiseRep* and baseline are trained alternately, and when training a part of the parameters, the rest of the parameters are frozen and fine-tuned for 10 epochs at a time, with a total number of epochs of 120. When evaluating, we average the results of the experiments under the same settings for 5 times, thus ensuring the reliability of the data.

## B  Experiment on Vehicle Identification

In the image retrieval task, we also conduct experiments to verify the effectiveness of our method on the vehicle recognition task. Vehicle recognition in practical scenarios often results in images containing a large amount of noise due to environmental factors such as lighting or occlusion, which increases the difficulty of detection. Our method is based on denoising to obtain features with better representation ability. Therefore, we want to experimentally verify whether the *DenoiseRep* plays a role in vehicle recognition tasks with higher noise levels. We select vehicleID [37] as the dataset, vehicle-ReID [11] as the baseline, and ResNet-50 as the feature extractor for the experiment.

Table 6: The performance of the *DenoiseRep* on vehicle recognition tasks.

| Method | Backbone | Datasets | mAP | Rank-1 |
|---|---|---|---|---|
| vehicle-ReID | ResNet-50 | vehicleID | 76.4% | 69.1% |
| vehicle-ReID + ***DenoiseRep*** | ResNet-50 | vehicleID | 77.3% | 70.2% |

From the results in Table 6, it can be seen that *DenoiseRep* demonstrates excellent performance in vehicle detection tasks. Compared to the baseline, adding the *DenoiseRep* significantly improves both mAP and Rank-1 metrics without incurring additional detection time costs. It verifies the denoising ability of the *DenoiseRep* in noisy environments.

## C  Experiment on Large Scale Image Classification Tasks

In this section, we aim to test the generalization ability of *DenoiseRep* in other tasks. We conduct experiments on two image classification datasets, ImageNet-1k and Cifar-10. These two datasets are both classic image classification datasets, rich in common images in daily life, and belong to large-scale image databases. ImageNet-1k is a subset of the ImageNet dataset, containing images from 1000 categories. Each category typically has hundreds to thousands of images, totaling over one million images. The Cfiar-10 contains 60000 32x32 pixel color images, divided into 10 categories. Each category contains 6000 images. To evaluate the effectiveness of our method, we use standard metrics, including Top-1 accuracy and Top-5 accuracy, which are commonly used to evaluate model performance in image classification tasks and are widely used on various image datasets, and we conduct detailed comparative experiments on multiple backbones and models with different parameter versions to verify the reliability of our method.

As shown in Table 7, we compare multiple classic backbones for representation learning on two datasets, and after adding the *DenoiseRep*, the accuracy metrics improves without adding model parameters. Our method demonstrates the capability to enhance the model's representation learning

Table 7: The effectiveness of our method in image classification tasks was validated on Cifar-10 and ImageNet-1k.

| Method | Datasets | Param | acc@1 | | acc@5 | |
|--------|----------|-------|-------|--|-------|--|
| | | | Baseline | +DenoiseRep | Baseline | +DenoiseRep |
| SwinV2-T [39] | ImageNet-1k | 28M | 81.82% | 82.13% | 95.88% | 96.06% |
| SwinV2-S [39] | ImageNet-1k | 50M | 83.73% | 83.97% | 96.62% | 96.86% |
| SwinV2-B [39] | ImageNet-1k | 88M | 84.20% | 84.31% | 96.93% | 97.06% |
| Vmanba-T [38] | ImageNet-1k | 30M | 82.38% | 82.51% | 95.80% | 95.89% |
| Vmanba-S [38] | ImageNet-1k | 50M | 83.12% | 83.27% | 96.04% | 96.22% |
| Vmanba-B [38] | ImageNet-1k | 89M | 83.83% | 83.91% | 96.55% | 96.70% |
| ViT-S [14] | ImageNet-1k | 22M | 83.87% | 84.02% | 96.73% | 96.86% |
| ViT-B [14] | ImageNet-1k | 86M | 84.53% | 84.64% | 97.15% | 97.23% |
| ResNet50 [18] | ImageNet-1k | 26M | 76.13% | 76.28% | 92.86% | 92.95% |
| ViT-S [14] | Cifar-10 | 22M | 96.13% | 96.20% | – | – |
| ViT-B [14] | Cifar-10 | 87M | 98.02% | 98.31% | – | – |

ability and extract more effective features through denoising, all while maintaining the same time costs. Moreover, *DenoiseRep* generalizes effectively to image classification tasks.

# D   Experiment on Image Detection Task

In this section, we aim to test the generalization ability of *DenoiseRep* in image detection tasks. We conduct experiments on the COCO [22] dataset. The COCO (Common Objects in Context) dataset is a widely used dataset for large-scale image recognition, object detection, and image segmentation, particularly in computer vision tasks. It contains 80 types of objects, such as people, animals, daily necessities, etc., covering various common items in daily life. To verify that our method is model independent, we conduct experiments using different models including Mask-RCNN [19], Faster-RCNN [49], ATSS [67], YOLO [48], DETR [6] and CenterNet [75], as well as diverse backbones. To evaluate the effectiveness of our method, we used standard metrics including AP, $AP_{50}$, and $AP_{75}$, which are commonly used metrics in object detection tasks to evaluate model performance, particularly for evaluating the effectiveness of bounding box detection. It is also an important part of the COCO dataset evaluation criteria, which can measure the detection ability of the model in multiple categories and scales.

Table 8: The effectiveness of our method in image detection tasks was validated on COCO.

| Methods | Backbones | AP | | $AP_{50}$ | | $AP_{75}$ | |
|---------|-----------|-----|--|-----------|--|-----------|--|
| | | Baseline | +DenoiseRep | Baseline | +DenoiseRep | Baseline | +DenoiseRep |
| Mask-RCNN | Swin-T | 42.8% | 44.3% | 65.1% | 67.1% | 47.0% | 48.6% |
| | Swin-S | 48.2% | 49.0% | 69.9% | 70.9% | 52.8% | 53.8% |
| | ResNet-50 | 42.6% | 43.2% | 63.7% | 65.0% | 46.4% | 46.8% |
| Faster-RCNN | ResNet-50 | 37.4% | 38.3% | 58.1% | 58.8% | 40.4% | 41.0% |
| ATSS | ResNet-50 | 39.4% | 39.9% | 57.6% | 58.2% | 42.8% | 43.2% |
| YOLO | DarkNet-53 | 27.9% | 28.4% | 49.2% | 50.3% | 28.3% | 27.8% |
| DETR | ResNet-50 | 39.9% | 40.8% | 60.4% | 59.9% | 41.7% | 42.9% |
| CenterNet | ResNet-50 | 40.2% | 40.6% | 58.3% | 59.1% | 43.9% | 44.0% |

As shown in Table 8, we compare multiple classic backbone networks across different methods. After adding *DenoiseRep*, the accuracy index shows improvement without the need for additional model parameters. This indicates that our method enhances the representation learning capability of the model and extracts more effective features through denoising, all while maintaining the same time costs. In addition, *DenoiseRep* well to generalized to image detection tasks.

# E    Experiment on Image Segmentation Task

In this section, our objective is to assess the generalization capability of *DenoiseRep* in image segmentation tasks. The image segmentation task aims to divide an image into multiple regions in order to identify and understand objects or areas within them. The main challenges it faces include: complex and varied backgrounds that can easily interfere with segmentation results, objects obstruct each other, making segmentation difficult, objects have diverse shapes and may undergo deformations, etc. We conduct experiments on the ADE20K [73] dataset using the current mainstream image segmentation models. ADE20K is a widely used scene segmentation dataset, mainly used for image segmentation tasks. It contains approximately 20000 images and over 150 different object and region categories. We choose mIoU and B-IOU as evaluation metrics to comprehensively evaluate the performance of image segmentation models. MIoU is the average of IoUs for all categories, which can effectively reflect the segmentation ability of the model on different categories. It provides a quantitative model for the accuracy of handling complex scenes by calculating the degree of overlap between the predicted area and the real area. A higher mIoU value means that the model can better identify and segment the target object. B-IoU focuses on evaluating the accuracy of segmentation boundaries and is particularly suitable for object edge segmentation tasks. It provides sensitivity to boundary details by measuring the degree of overlap between predicted boundaries and real boundaries.

Table 9: The effectiveness of our method in image segmentation tasks was validated on ADE20K.

| Methods | Backbones | aAcc | | B-IoU | | mIoU | |
|---|---|---|---|---|---|---|---|
| | | *Baseline* | *+DenoiseRep* | *Baseline* | *+DenoiseRep* | *Baseline* | *+DenoiseRep* |
| FCN [40] | ResNet-50 | 0.774 | 0.779 | 0.287 | 0.299 | 0.359 | 0.365 |
| FCN | ResNet-101 | 0.793 | 0.796 | 0.306 | 0.316 | 0.396 | 0.404 |
| SegFormer [63] | mit_b0 | 0.782 | 0.788 | 0.292 | 0.297 | 0.374 | 0.381 |
| SegFormer | mit_b1 | 0.812 | 0.816 | 0.341 | 0.348 | 0.422 | 0.425 |

As shown in Table 9, we compare two classical backbone networks with different methods. After adding the *DenoiseRep*, both IoU metrics improve without adding model parameters. Practice proves that our method can improve the representation learning ability of the model and obtain more effective features through denoising without increasing additional time costs. In addition, *DenoiseRep* well to generalized to image segmentation tasks.

# F    Fairness Experiment

To ensure the fairness of the experiment, we compared the performance of the baseline method with our proposed method under the same conditions. Specifically, during the training process, we strictly controlled the experimental variables so that both the baseline method and our method ran under the same number of training epochs and hyperparameter settings.

Table 10: Comparison of performance between baseline method and our proposed method in the same additional training epochs. The baseline method TransReID-SSL is based on ViT-small backbone.

| Epoch | 120 | 160 | 200 | 240 |
|---|---|---|---|---|
| Baseline | 81.18% | 81.19% | 81.18% | 81.16% |
| *+DenoiseRep* | 81.18% | 81.64% | 82.00% | 82.12% |

The experimental results in Table 10 indicate that the baseline method did not show significant performance improvement under the same number of training epochs. This observation indicates that the performance improvement obtained is attributed to our proposed method, rather than an increase in training time or number of epochs, which validates the effectiveness of our method.

# G    Limitations

Our proposed method *DenoiseRep* improves the accuracy of current mainstream backbones while ensuring label-free and no additional computational costs, and it has been experimentally verified to

be generalizable in multiple image tasks. However, from the experimental results, it can be found that our method has limited improvement in model accuracy when generalized to general tasks, and in order to fuse the parameters of the denoising layer and the feature extraction layer, only one or two steps of denoise for each denoising layer, and the number of denoising layers is not more than that of the feature extraction layer, which limits the denoising intensity. We will continue to explore how to further improve the accuracy of the model without adding additional inference time costs or only adding a small number of additional parameters.

