# OpenReview forum: "DenoiseRep: Denoising Model for Representation Learning"
_NeurIPS.cc/2024/Conference — NeurIPS 2024 oral_

### Official Review · Reviewer_Ckqo · 2024-07-08

**Soundness:** 4
**Presentation:** 4
**Contribution:** 4
**Rating:** 8
**Confidence:** 5

**Summary:**

This paper proposes DenoiseReID to improve feature discriminative with joint feature extraction and denoising, in which FEFDFA is developed to merge parameters of the denoising layers into embedding layers. Experimental results show the proposed DenoiseReID improves performance.

**Strengths:**

1.	The proposed joint representation learning and denoising process.
2.	The proposed FEFDFA is a computation-efficient algorithm.

**Weaknesses:**

The author solves all my quentions.

**Questions:**

N/A

**Limitations:**

Yes

---

> ### Author Rebuttal · Authors · 2024-08-06
>
> Q1: **What constitutes this noise?**
>
> A1: We appreciate your detailed review. Experimental results of Table 3 (FEFDUF) could empirically demonstrated the hypothesis of "the features obtained by backbone extraction are noisy". FEFDUF includes a well-trained person ReID model, and a denoise model which take feature of the ReID model as input and predict its noise. During the training stage, the ReID model is always frozen, and denoise model is trained under normal process in DDPM (i.e. adding a gaussian noise to feature, taking the nosied feature as input and predicting the original gaussian noise). During the inference stage, given images, we first extract features A with the well-trained ReID model, then put the features into the denoise module to predict its noise A'. Finally, we found that A-A' performs stably better than A. We will add the analysis above to the revised manuscript.
> &emsp;
>
> Q2: **The authors should evaluate the proposed method on these datasets instead of conventional ones.**
>
> A2: Thank you for the valuable suggestion. We validate our proposed FEFDFA on Occluded-Duke [1], Occluded-ReID [2] and Partial-ReID [3]. These datasets include person images occluded by misc obstacles such as vehicles, trees and so on, making them more challenging. We will add the experimental results to the revised manuscript. The experimental results are as follows, using mAP as the evaluation metric:
>  |   Dataset    |  P-ReID |  OCC_Duke | OCC-ReID |
>  |:-------|:------------:|:------------:|:------------:|
>  | TransReID |  72.3%    | 59.5%    | 71.2%    |
>  | TransReID+FEFDFA   |  73.5% (↑1.2%) |   60.4%  | 72.0% |
>
> Q3: **The authors lack an analysis of the model's complexity, including the number of parameters and training time.**
>
> A3: Thanks for your valuable feedback. Our proposed method takes LITTLE extra parameters and training time. The reason is the denoising layers employ simple linear layers, they are very light compared with heavy vision backbone (e.g. ViT, ResNet). For example, the parameter of ViT-Small is 22,313,320, while the sum of the parameters of the denoising module is only 3,538,944, an additional increase of about 15.8%. The detailed experimental results are as follows. Please note that these are training-time parameters, our proposed method takes no extra latency during the test stage.
>  |   Backbone    |  Baseline |  Ours | Increase |
>  |:-------|:------------:|:------------:|:------------:|
>  | ViT-Small |  22,313,320    | 25,852,264    | 15.8%    |
>  | ViT-Base   |  89,781,544 |   103,937,320 | 11.6% |
>
>
> Q4: **The performance gain are minimal.**
>
> A4: Thanks for your valuable feedback. Our proposed method is test-time computation-free (FREE LAUNCH!) and scalable to various baseline and downstream tasks, which is proven to be effective in CNN series (e.g. ResNet50), Transformer series (e.g. ViT, Vmanba), Person Re-ID (MSMT, Duke), Vehicle Re-ID (vehicReID), Image Classification (ImageNet, CUB). The latest experimental results on CLIP-REID show better improvement on MSMT (CLIP-REID: 75.8% v.s. CLIP-REID+OURS: 76.5% for mAP, see the table below). The code will be released after the manuscript is accepted. We will add the experimental results to the revised manuscript.
> &emsp;
>
> Q5: **The authors should verify the effectiveness of the proposed method using the visual encoder in CLIP as the backbone.**
>
> A5: Thanks for your insightful suggestion. CLIP is a very strong vision-text encoder, which is trained with 400 million data. Beyond CLIP, CLIP-ReID pioneeringly adapts CLIP, a zero-shot classifier, to ReID, a fine-grained image retrieval task. Our proposed FEFDFA is model-free, thus it can be easily applied to CLIP-ReID without any modification. The experimental results on CLIP-REID show stable improvement on MSMT (CLIP-REID: 75.8% v.s. CLIP-REID+OURS: 76.5% for mAP). The code will be released if the manuscript could be accepted. We will add the experimental results to the revised manuscript. The experimental results are as follows, with mAP as the evaluation metric:
>
>  |   Dataset    |  DukeMTMC |  MSMT | Market-1501 |
>  |:-------|:------------:|:------------:|:------------:|
>  | CLIP-REID |  83.1%    | 75.8%    | 90.5%    |
>  | CLIP-REID+FEFDFA   |  83.9%(↑0.8%) |   76.5%(↑0.7%) | 91.1% |
>
>
>  [1] Jiaxu Miao, Yu Wu, Ping Liu, Yuhang Ding, and Yi Yang. Pose-guided feature alignment for occluded person re-identification. In ICCV, pages 542–551, 2019. 1, 2, 6
>
>  [2] Jiaxuan Zhuo, Zeyu Chen, Jianhuang Lai, and Guangcong Wang. Occluded person re-identification. In ICME, pages 1–6. IEEE, 2018. 2, 6
>
> [3] Wei-Shi Zheng, Xiang Li, Tao Xiang, Shengcai Liao, Jianhuang Lai, and Shaogang Gong. Partial person reidentification. In ICCV, pages 4678–4686, 2015. 6

---

> ### Author Response · Authors · 2024-08-12
> **Comment by Authors**
>
> Hi, dear reviewer, we hope our responses have solved your concerns. If you still have some concerns, please feel free to raise them here and we will respond as soon as possible.

---

> ### Comment · Reviewer_Ckqo · 2024-08-12
>
> The authors slove all my quentions. Overall, the motivation of this paper is clear, the idea is novel, the proposed is simple yet effective, and the writting is satifactory.

---

> > ### Author Response · Authors · 2024-08-12
> > **Responses by Authors**
> >
> > We thank the reviewer kind comment. We will keep improving the proposed method and apply it to more downstream tasks in future.

---

### Official Review · Reviewer_DSeM · 2024-07-09

**Soundness:** 3
**Presentation:** 3
**Contribution:** 3
**Rating:** 8
**Confidence:** 5

**Summary:**

This mauscript proposes a novel denosing model for representaetion learning and take person re-identification as a benchmark. It unifies the frameworks of feature extraction and feature denoising, where the former progressively embeds features from lowlevel to high-level, and the latter recursively denoises features step-by-step. Besides, a FEFDA is proposed to fuse feature extraction and denoising in a single backbone without changing its structure and taking extra runtime latency. Experiments on ReID, large-scale  and fine-grained classification tasks show its effectivenss.

**Strengths:**

1. The idea of unifying feature extraction and feature denoising in a single backbone without changing its structure is interesting. As far as I know, its first time to see the idea.
2. The characteristic computation-free and label-free is promising. The thoerical analysis seems precies and right.
3. Experiments on 3 typical tasks and 9 datasets of representation learning (retrieval, classfication, fine-grained classification) are sufficient and extensive.

**Weaknesses:**

1. the proposed "Feature Extraction and Feature Denoising Fusion Algorithm" is little similar to reparameterization, please clarify their difference.
2. its application to transformer series are well analyzed, it will be better if peformance on CNN series are shown.
3. representation learning is a wide and fundational conception, its applications on more downstream tasks, such as detection, segmentation even generation, could be analyzed,  in future.
4. The hyper-parameter analysis is missed.
5. This method seems need extra training steps, what if the baseline methods are trained under the same steps?

**Questions:**

For questions, please see the weakness in the above section.

**Limitations:**

The authors have discussed the limitations of the proposed method.

---

> ### Author Rebuttal · Authors · 2024-08-06
>
> Q1: **Clarify the difference between the proposed FEFDFA and reparameterization.**
>
> A1: Thank you for your insightful comment. We appreciate the opportunity to clarify the differences between our "Feature Extraction and Feature Denoising Fusion Algorithm" (FEFDFA) and reparameterization.
>    - Reparameterization: Reparameterization typically involves restructuring the parameters of a model to facilitate more efficient training. It is widely used in variational autoencoders (VAEs) to allow for gradient-based optimization of stochastic variables.
>    - Feature Extraction and Feature Denoising Fusion Algorithm (FEFDFA): Our FEFDFA is designed to integrate feature extraction and denoising within the same backbone. Each embedding layer in the backbone serves a dual purpose: extracting features and performing denoising simultaneously. By deriving the formula for diffusion model sampling, the denoising layer parameters are fused with the feature layer parameters, so that the model has no additional time cost in the inference stage.
> &emsp;
>
> Q2: **It will be better if peformance on CNN series are shown.**
>
> A2: Thanks for your valuable feedback. Our proposed method is scalable to various baseline (including CNN, see Table 5 and line seven of Table 6 for details)  and downstream tasks, including CNN series (e.g. ResNet50), Transformer series (e.g. ViT, Vmanba), Person Re-ID (MSMT, Duke), Vehicle Re-ID (vehicReID), Image Classification (ImageNet, CUB). The experimental results in Section 4.4 of the main text and Appendix C demonstrate that our method significantly improves these tasks.
> &emsp;
>
> Q3: **Representation learning applications on more downstream tasks.**
>
> A3: Thanks for your valuable suggestion. Representation learning is a broad and foundational concept with potential applications across various downstream tasks. Due to the limited time, we extend our proposed FEFDFA to the object detection task with Mask-RCNN[2] as a baseline and COCO[1] as a benchmark. Experimental results show that ours carries stable improvements of 1.6%-1.1%. Please see the table below for details. We will extend ours to more downstream tasks including segmentation and generation, in future.
>
>  | **MaskRCNN** | **bbox_mAP** | **bbox_mAP_50** | **bbox_mAP_75** | **mask AP** |
>  |--------------|:--------------:|:-----------------:|:-----------------:|:-------------:|
>  | Swin-T       | 0.427        | 0.652           | 0.468           | 0.393       |
>  | Swin-T+FEFDFA | 0.443(↑1.6%)       | 0.671           | 0.486           | 0.405       |
>  | Swin-S       | 0.482        | 0.698           | 0.528           | 0.432       |
>  | Swin-S+FEFDFA | 0.493(↑1.1%)       | 0.709           | 0.540           | 0.439       |
>
> Q4: **The hyper-parameter analysis is missed.**
>
> A4: Thanks for your reminder. We analyze hyperparameters in the model, including important parameters such as $\beta_t$ (Line178), and diffusion step size T (Line73). The results of the hyper-parameter analysis will be included in the revised manuscript, along with discussions on how different settings affect the model's performance. We use ViT-small as the backbone and conducted experiments on the DukeMTMC dataset with mAP as the evaluation metric. The experimental results are as follows:
>  | **$\beta_t$** | **[1e-3, 0.02]** | **[1e-4, 0.02]** | **[1e-5, 0.02]** | **[1e-6, 0.02]** |
>  |:--------------:|:--------------:|:-----------------:|:-----------------:|:-------------:|
>  | mAP       | 81.15        | 81.22           | 81.21           | 81.13      |
>
>  | **T** |  **100** | **500** | **1000** | **2000** | **5000** |
>  |:------:|:------:|:-------:|:-------:|:-------:|:-------:|
>  | mAP   |  80.92  | 81.15   | 81.22   | 81.19   | 80.98   |
>
> Q5:**What if the baseline methods are trained under the same steps?**
>
> A5: Thank you for your question. We conduct experiments to address this concern. We train the baseline methods using the same number of steps as our proposed method. However, we do not observe any significant performance improvements in the baseline methods. We will include these experimental results and detailed analyses in the revised manuscript to support this explanation. Thank you for your valuable feedback. The experimental results are as follows：
>
>  |   epoch    |  120 |  160 |  200 |  240 |
>  |:-------|:------------:|:------------:|:------------:|:----------:|
>  | Baseline | 80.38%    | 80.39%    | 80.38%    | 80.36%   |
>  | Ours   | 80.38% |  80.84% | 81.20% | 81.22% |
>
>    where the original baseline trained for 120 epochs. Therefore, when the epoch is 120, our method did not participate in fine-tuning. When the epoch is 160, our method fine-tunes for another 40 epochs based on the pre-trained parameters of the baseline, and so on.
>
> [1] T.-Y. Lin, M. Maire, S. Belongie, J. Hays, P. Perona, D. Ramanan, P. Dollar, and C. L. Zitnick. Microsoft COCO: Common objects in context. In ECCV, 2014. 2, 5
>
>  [2] He, Kaiming, et al. "Mask r-cnn." Proceedings of the IEEE international conference on computer vision. 2017.

---

> > ### Comment · Reviewer_DSeM · 2024-08-09
> >
> > Thanks for authors's feedback. After reading it, all of my questions have been addressed.
> >
> > I also read the comments from other reviewers and notice an extra point "minimal gains on TransReID".  The authors claim that 1.1-1.6% extra gain based on MaskRCNN and 0.6-0.8% extra gain based on latest state-of-the-art CLIP-ReID. I thought this is a satisfying improvement.
> >
> > I prefer its novelty, concision, and its advantages of no-extra-lancy, stable improvements and scalibity to many downstream tasks. In summary, I keep my original rating (strong accept). Looking forward to seeing its application to more tasks.

---

> > > ### Author Response · Authors · 2024-08-11
> > > **Reponse to reviewer's comment**
> > >
> > > We thank the reviewer kind comment. We will keep improving the proposed method and apply it to more downstream tasks in future.

---

### Official Review · Reviewer_4MBF · 2024-07-11

**Soundness:** 2
**Presentation:** 3
**Contribution:** 2
**Rating:** 6
**Confidence:** 5

**Summary:**

This paper proposes a new method, Feature Extraction and Feature Denoising Fusion Algorithm (FEFDFA), which utilizes the denoising ability of diffusion models to denoise the features in the feature extraction layer, and fuses the parameters of the denoising layer with those of the feature extraction layer through parameter fusion, further improving retrieval accuracy without incurring additional computational costs. The effectiveness of the FEFDFA method has been validated on multiple common image discrimination task datasets.

**Strengths:**

1.The article structure is complete and writing is generally clear.

2.Some experimental results seem to good.

**Weaknesses:**

1.In fact, the intermediate layer features of the pre-trained diffusion model can be used directly for the downstream task such as the discrimination task [1][2][3].
The authors need to enrich the Related Work.

2.Line144-146, no evidence provided to support the proposed hypothesis.

3.The authors treat the backbone as a series of denoising layers, so the training loss in Eq. (11) should be the sum of the MSE losses of each denoising layer.

4.The statement is inconsistent.

Line172-173，"we freeze the original parameters and only trained the FEFDFA."

Line509-511，"the parameters of the FEFDFA and baseline were trained alternately."

In addition, unless the baseline and FEFDFA are trained together, Eq. (12) is incorrect.


[1] Mukhopadhyay, Soumik, et al. "Diffusion models beat gans on image classification." arXiv preprint arXiv:2307.08702 (2023).

[2] Li, Alexander C., et al. "Your diffusion model is secretly a zero-shot classifier." ICCV 2023.

[3] Baranchuk, Dmitry, et al. "Label-Efficient Semantic Segmentation with Diffusion Models." ICLR 2022.

**Questions:**

same to the weaknesses.

**Limitations:**

The authors have addressed the limitations.

---

> ### Author Rebuttal · Authors · 2024-08-06
>
> Q1: **The authors need to enrich the Related Work.**
>
> A1: We thank the valuable suggestions. Our proposed DenoiseReID is different from the related works [1-3]. The related works [1-3] apply the itermediate layer features of an existing pre-trained diffusion model to improve downstream task. Ours applies the denoise algorithm to any existing pre-trained downstream model (e.g. Person ReID of Table 2, Vehicle ReID of table 5, Image Classification of Table 6). Compared with the related works [1-3], ours is more scalable (suitable to more downstream tasks) and light-weight (computation-free). We will carefully review and add them to the revised manuscript.
>   &emsp;
>
> Q2: **Line144-146, no evidence provided to support the proposed hypothesis.**
>
> A2: Thanks for the kind comment. Experimental results of Table 3 (FEFDUF) could empirically demonstrate the hypothesis of "the features obtained by backbone extraction are noisy". FEFDUF includes a well-trained person ReID model, and a denoise model which takes the feature of the ReID model as input and predicts its noise. During the training stage, the ReID model is always frozen, and the denoise model is trained like DDPM (i.e. adding a gaussian noise to feature, taking the nosied feature as input and predicting the original gaussian noise). During the inference stage, given images, we first extract feature A with the well-trained ReID model, then put the feature into the denoise module to predict its noise A'. Finally, we found that A-A' performs better than A. The training process DO NOT use any label because DDPM is an unsupervised manner. This experiment shows that simple denoising features contribute to improvement, partially supporting the view that "the features obtained by backbone extraction are noisy". We will add the analysis above to the revised manuscript.
> &emsp;
>
> Q3: **The training loss in Eq. (11) should be the sum of the MSE losses of each denoising layer.**
>
> A3: Thanks for the valuable reminder. This is a writing error. In the code implementation, the loss during the training of the denoising module is the sum of the MSE losses of each denoising layer. The code will be released if the manuscript could be accepted. We will polish the Eq. (12) to be:
> $$
> Loss_p = \sum_{i=1}^{N} \left\| \epsilon_i - D_{\theta_i} (X_{t_i}, t_i) \right\|
> $$
>
> Q4: **The statement is inconsistent.**
>
> A4: Thanks for your comments. There are THREE questions. Let's discuss them ONE BY ONE:
>   - (1) "we freeze the original parameters and only train the FEFDFA." This is our basic training setting. It only trains the parameters of FEFDFA with denoising and (optional) ReID loss. In all unsupervised settings, we use this training set (without ReID loss). It makes significant improvements. Specifically, in the experimental results between the second and first lines of Table 1, it can be observed that the performance of the unsupervised FEFDFA method has significantly improved compared to the baseline.
>   &emsp;
>   - (2) "the parameters of the FEFDFA and baseline were trained alternately."  Alternately training is a TRAINING TRICK for the supervised setting. Specifically, we (a) freeze the original parameters, train FEFDFA with denoising and reid losses, then (b) merge parameters of FEFDFA into the original parameters, train latest original parameters with ReID loss only, (c) repeat (a) and (b). In all supervised settings, we use this trick. It carries more improvement. Specifically, the experimental results in the second and third lines of Table 1. Note that step (a) still carries improvements in the supervised setting.
>   &emsp;
>   - (3) "In addition, unless the baseline and FEFDFA are trained together, Eq. (12) is incorrect." Eq.(12) is CORRECT even if parameters of FEFDFA are trained and the original parameters are frozen. Both ReID loss and denoising losses can be used to optimize parameters of FEFDFA. This equals to (2)-(a) above. Similar ideas of "using task-related loss to supervise denoising module" can be found in DiffDet[4] and DiffSeg[5].
>
>
> [1] Mukhopadhyay, Soumik, et al. "Diffusion models beat gans on image classification." arXiv preprint arXiv:2307.08702 (2023).
>
> [2] Li, Alexander C., et al. "Your diffusion model is secretly a zero-shot classifier." ICCV 2023.
>
> [3] Baranchuk, Dmitry, et al. "Label-Efficient Semantic Segmentation with Diffusion Models." ICLR 2022.
>
> [4] Chen, Shoufa, et al. "Diffusiondet: Diffusion model for object detection." Proceedings of the IEEE/CVF international conference on computer vision. 2023.
>
> [5] Tian, Junjiao, et al. "Diffuse Attend and Segment: Unsupervised Zero-Shot Segmentation using Stable Diffusion." Proceedings of the IEEE/CVF Conference on Computer Vision and Pattern Recognition. 2024.

---

> > ### Comment · Reviewer_4MBF · 2024-08-10
> >
> > Thanks for the authors' feedback. I am still concerned about the following issues.
> >
> > For Q1, I think the motivation of this paper is the same as that of [1, 2, 3], both of which apply denoising features to downstream tasks. I think it makes more reasonable to directly use the features of the pre-trained diffusion model.
> >
> > For Q2, the results in Table 3 cannot provide strong evidence for the hypothesis stated by the authors in Line144-146 of the paper. Since the ReID loss is used to optimize the parameters of FEFDFA and the parameters of the FEFDFA are fused with the backbone in the final, I suspect that the performance improvement in the Table 3 is more like the data augmentation effect brought about by adding noise, rather than denoising.
> >
> > For Q4, unless the authors provide relevant theory or previous works to prove that denoising loss can be used with other task losses to optimize diffusion models or visual task models, I believe that the training loss Eq.(12) is questionable. Denoising in diffusion is fundamentally different from other downstream visual tasks as the optimization objective of them are inconsistent.
> > The relevant paper listed by the author cannot provide evidence to the rebuttal.
> > DiffDet[4] applies the idea of denoising to the bbox regression in object detection.
> > DiffSeg[5] directly uses the intermediate attention layer features of the stable diffusion for the segmentation task, which is similar to the method [1,2,3] mentioned in Q1 that directly uses the intermediate layer features of the pre-trained diffusion model.
> >
> > [1] Mukhopadhyay, Soumik, et al. "Diffusion models beat gans on image classification." arXiv preprint arXiv:2307.08702 (2023).
> >
> > [2] Li, Alexander C., et al. "Your diffusion model is secretly a zero-shot classifier." ICCV 2023.
> >
> > [3] Baranchuk, Dmitry, et al. "Label-Efficient Semantic Segmentation with Diffusion Models." ICLR 2022.
> >
> > [4] Chen, Shoufa, et al. "Diffusiondet: Diffusion model for object detection." ICCV 2023.
> >
> > [5] Tian, Junjiao, et al. "Diffuse Attend and Segment: Unsupervised Zero-Shot Segmentation using Stable Diffusion." CVPR 2024.

---

> > > ### Author Response · Authors · 2024-08-11
> > > **Response to Comment by Reviewer 4MBF**
> > >
> > > We thank the reviewer's detailed comment. It seems that there are still some misunderstandings. Please allow us to explain them again. The responses are listed below and free feel to deeply discuss.
> > >
> > > **Q: "For Q2"**
> > >
> > > A:  As we have clarified in the rebuttal, we DO NOT use any label (i.e. ReID loss is NOT used) in this experiment. The improvement should come from denoising loss. Based on the observation, we "suppose that in the inference stage, the features obtained by backbone extraction are noisy".
> > >
> > > **Q: "For Q4"**
> > >
> > > A: As far as I know, "unifying feature extraction and feature denoising" is pioneeringly proposed in this manuscript. Thus, we can't provide previous works to prove "why denoising loss and task losses can be used together". However, we show the fake code of our proposed method, hoping this can solve your misunderstanding. Please pay attention to the line "denoised_feats = feats - self.denoise_layer(x)", which we think may cause your misunderstanding.
> > >
> > > ```
> > > // Fake Code of the Proposed Algorithm
> > > // Plz note that, this code targets understanding "why reid_loss and denosing_loss can be used together". Thus, we only show some critical logic and details may be missed and inaccurate.
> > >
> > > class DenoiseLinear:
> > >
> > > 	def __init__(self):
> > > 		self.linear = nn.Linear()
> > > 		self.denoise_layer = nn.Linear()
> > >
> > > 		set_require_grad_false(self.linear)
> > >
> > > 	def forward_train(self, x):
> > > 		// basic branch
> > > 		feats = self.linear(x)
> > >
> > > 		// denoising branch, this process is the same with DDPM
> > > 		gt_noise = sample_from_gaussian()
> > > 		pred_noise = self.denoise_layer(x + gt_noise)
> > > 		denoise_loss = l1_or_l2_loss(gt_noise,  pred_noise)
> > >
> > > 		// tip to optimzie denoise layer with reid_loss is returning denoised_feats NOT feats
> > > 		denoised_feats = feats - self.denoise_layer(x)
> > > 		return denoised_feats, denoise_loss
> > >
> > > 	def forward_test(self, x);
> > > 		w, b = fuse_weight(self.linear, self.denoise_layer) // compute offline
> > > 		denosied_feats = w * x + b
> > > 		return denosied_feats
> > >
> > >
> > > // A Toy ReID Model with 2 linear layers
> > > class ReIDModel:
> > > 	def __init__(self):
> > > 		self.linear1 = DenoiseLinear()
> > > 		self.linear2 = DenoiseLienar()
> > >
> > >   // unsupervised manner
> > > 	def train_without_reid_loss(self, x):
> > > 		feat1, denoise_loss1 = self.linear1.forward_train(x)
> > > 		feat2, denoise_loss2 = self.lienar2.forward_train(x)
> > > 		return feat2, denoise_loss1 + denoise_loss2
> > >
> > > 	// supervised manner
> > > 	def train_with_reid_loss(self, x, y):
> > > 		feat1, denoise_loss1 = self.linear1.forward_train(x)
> > > 		feat2, denoise_loss2 = self.lienar2.forward_train(x)
> > > 		reid_loss = compute_reid_loss(feat2, y)
> > > 		return feat2, denoise_loss1 + denoise_loss2 + reid_loss
> > >
> > > 	def forward_test(self, x):
> > > 		feat1 = self.linear1.forward_test(x)
> > > 		feat2 = self.linear2.forward_test(feat1)
> > > 		return feat2
> > >
> > > ```
> > >
> > > **Q: "For Q1"**
> > >
> > > A: Our proposed method is very different from the related work [1-3]. We summarize them in the table below:
> > >
> > > |                     | Usage                                                        | Scalibity                                                    | Improvement and Latency                                      |
> > > | :-----------------: | ------------------------------------------------------------ | ------------------------------------------------------------ | ------------------------------------------------------------ |
> > > |        Ours         | Improve existing vision models, which could be from many vision tasks. | [better] ONE method for MANY vision taks without customizing implementation and hyper-parameters. | [better] Given a model of specific vision task (e.g. CLIP-REID), achieve STABLE improvement with NO extra latency compared to the given model. |
> > > | Related Works [1-3] | Customize a model for a specific vision task based on a well-trained strong diffusion model. | ONE method for ONE vision task. Need customize Implementation and hyper-parameters for specific tasks. | Given a diffusion model (e.g. StableDiffusionXL),  improvement and latency depend on how strong the diffusion model is and the details of the customized implementation for specific tasks. |

---

> > > > ### Comment · Reviewer_4MBF · 2024-08-13
> > > >
> > > > Thank you for your the positive response. The authors have addressed some of my concerns. I will revise my initial rating, combined with the comments of other reviewers.

---

> > > > > ### Author Response · Authors · 2024-08-13
> > > > > **Response by Authors**
> > > > >
> > > > > Thank you for your kind update. Should you have any additional comments or questions, please feel free to share them with us. We are more than happy to respond to them.

---

### Official Review · Reviewer_w7Th · 2024-07-12

**Soundness:** 3
**Presentation:** 3
**Contribution:** 3
**Rating:** 8
**Confidence:** 4

**Summary:**

This paper proposes a novel denoising model called DenoiseReID, designed to enhance representation learning in person re-identification (ReID) tasks. This approach combines traditional denoising processes with feature extraction through a feature extraction and denoising fusion algorithm (FEFDFA) that incurs no additional computational cost. It incrementally improves the discriminability of features without the need for labels.

**Strengths:**

The experimental results demonstrate that DenoiseReID achieves stable performance improvements across multiple ReID datasets. Furthermore, it can be extended to large-scale image classification tasks such as ImageNet, CUB200, Oxford-Pet, and Flowers datasets.

**Weaknesses:**

1. In the comparative experiments, the authors missed an opportunity to benchmark their method against the latest advancements like CLIP-ReID, restricting comparisons solely to TransReID.

2. When contrasted with TransReID, this approach shows a moderate enhancement in performance.

**Questions:**

I am particularly concerned about the performance comparison in this paper. Is it necessary to use diffusion models for conducting ReID tasks, and what advantages do they offer compared to CNNs or ViTs? Currently, it seems that their performance improvement is rather modest.

**Limitations:**

yes, the authors adequately addressed the limitations.

---

> ### Author Rebuttal · Authors · 2024-08-06
>
> Q1: **The authors missed an opportunity to benchmark their method against the latest advancements like CLIP-ReID.**
>
> A1: Thanks for your valuable feedback. CLIP is a very strong vision-text encoder, which is trained with 400 million data. Beyond CLIP, CLIP-ReID pioneeringly adapts CLIP, a zero-shot classifier, to ReID, a fine-grained image retrieval task. Our proposed method is model-free, thus it can be easily applied to CLIP-ReID without any modification. The experimental results of CLIP-ReID show stable improvement on the three datasets(DukeMTMC, MSMT, Market-1501). The code will be released if the manuscript could be accepted. We will add the experimental results to the revised manuscript. The experimental results are as follows, with mAP as the evaluation metric:
>
> |   Dataset    |  DukeMTMC |  MSMT | Market-1501 |
> |:-------|:------------:|:------------:|:------------:|
> | CLIP-REID |  83.1%    | 75.8%    | 90.5%    |
> | CLIP-REID+FEFDFA   |  83.9%(↑0.8%) |   76.5%(↑0.7%) | 91.1% |
>
> Q2: **When contrasted with TransReID, this approach shows a moderate enhancement in performance.**
>
> A2: Thanks for your kind comment.  Our proposed method is test-time computation-free (FREE LAUNCH!) and scalable to various baseline and downstream tasks, which is proven to be effective in CNN series (e.g. ResNet50), Transformer series (e.g. ViT, Vmanba), Person Re-ID (MSMT, Duke), Vehicle Re-ID (vehicleID), and Image Classification (ImageNet, CUB). For its scalability, we DO NOT hack it for backbones or tasks. With the same hyper-parameters, ours achieves consistent and stable improvement on ALL backbones and tasks above. For example, based on previous state-of-the-art CLIP-ReID, our proposed method achieves extra improvement from 0.8%-0.6% on Duke, MSMT and Market.
>
> Q3: **Is it necessary to use diffusion models for conducting ReID tasks, and what advantages do they offer compared to CNNs or ViTs?**
>
> A3: Thanks for your kind comment. In this work, we use the ViT series for conducting ReID tasks. Our proposed Feature Extraction and Feature Denoising Fusion Algorithm (FEFDFA) fuse the idea of diffusion into the backbone (e.g. ResNet, ViT). We treat each block of the backbone as a denoising layer and utilize the denoising ability of the diffusion model to denoise on a different feature extraction level. During the inference phase, the denoising layer parameters are fused with the backbone network parameters without incurring additional inference time cost.

---

> > ### Comment · Reviewer_w7Th · 2024-08-10
> >
> > After carefully reviewing the author's rebuttal, I believe that the concerns I previously raised have been thoroughly addressed. I have decided to accept this paper for the following main reasons:
> >
> > This paper presents a novel representation learning denoising model for person re-identification, named DenoiseRelD. By integrating feature extraction and denoising techniques, the model significantly enhances feature discriminability. The approach is both theoretically elegant and practically feasible. The author has clearly articulated the motivation and design details of the DenoiseRelD method in both the paper and the rebuttal. Compared to existing methods, DenoiseRelD demonstrates outstanding performance across four benchmark datasets, showing significant advantages.
> >
> > Given these reasons, along with the author's effective response to the concerns raised by other reviewers, I have decided to revise my initial rating to  'Strong Accept'.

---

> > > ### Author Response · Authors · 2024-08-11
> > > **Response to reviewer's comment**
> > >
> > > We thank the reviewer kind comment. We will keep improving the proposed method and apply it to more downstream tasks in future.

---

### Author Response · Authors · 2024-08-06
**Open Discussion**

Hi dear reviewers, we thank all your professional and kind comments. We have carefully read and responded to all the comments. Please allow me to clarify some misunderstandings and highlight our contributions.

Our proposed method is test-time computation-free (FREE LUNCH!) and scalable to various baseline and downstream tasks, which has been proven in CNN series (e.g. ResNet50), Transformer series (e.g. ViT, Vmanba), Person Re-ID (MSMT, Duke), Vehicle Re-ID (vehicReID), Large-Scale Image Classification (ImageNet), Fine-Grained Image Classification (CUB) and Object Detection (COCO). For its scalability, we DO NOT hack it for backbones or tasks. With the same hyper-parameters, ours achieve consistent and stable improvement on ALL backbones and tasks above.

For example, based on the famous object detection method Mask-RCNN, our proposed method achieves extra improvement from 1.6%-1.1% on COCO. Based on previous state-of-the-art CLIP-ReID, our proposed method achieves extra improvements from 0.8%-0.6% on Duke, MSMT and Market.  All those improvements take NO extra test-time computation and NO specific hyper-parameters customized by tasks.

Considering the advantages of **test-time computation-free** and the **consistent and stable improvement on the MANY backbones and tasks** (CNN series, Transformer series, Person ReID, Vehicle ReID, Occluded ReID, Large-Scale Image Classification, Fine-Grained Image Classification, Object Detection), we believe the proposed method will make significant contributions to the Representation Learning community.

Code will be released in future.

---

### Decision · Program_Chairs · 2024-09-25

**Decision:**

Accept (oral)

**Comment:**

The paper received all accept recommendations. Reviewers found the proposed method novel and interesting. AC agrees with reviewers and is happy to accept this paper for publication.